# Occupational Exposures and Esophageal Cancer: Prog Study

**DOI:** 10.3390/ijerph19169782

**Published:** 2022-08-09

**Authors:** Annabelle Gressier, Greta Gourier, Jean-Philippe Metges, Jean-Dominique Dewitte, Brice Loddé, David Lucas

**Affiliations:** 1Occupational and Environmental Diseases Center, Teaching Hospital, F-29200 Brest, France; 2Cancerology and Hematology Institute, Teaching Hospital, F-29200 Brest, France; 3ORPHY Laboratory, University Brest, F-29200 Brest, France; 4Seamen’s Health Service, Ministry of Transport, F-92040 Paris, France

**Keywords:** esophageal cancer, occupational exposure, observational study

## Abstract

Esophageal cancer is the sixth most common cause of cancer death worldwide. In France, Brittany is one of the regions most seriously affected. This increased incidence is usually linked to high rates of alcohol overconsumption and smoking, established risk factors for esophageal cancer, but the region has special occupational exposures. We aim to describe the occupational exposures of patients with esophageal cancer. Between June and October 2020, we conducted a monocentric descriptive study in a French Teaching Hospital and identified 37 eligible patients. We gathered data through a systematic individual interview for each participant and by an analysis of their medical file. We were able to include 36 patients; most were men (n = 27, 75.0%) and smokers (n = 25, 69.4%), 21 (58.3%) presented an adenocarcinoma esophageal cancer, 13 (36.1%) a squamous cell cancer, and 2 other types. On occupational exposure, patients declared respectively high exposure by manipulating asbestos materials for 11 (30.6%) patients, regularly in contact with benzene by handling fuel in 7 cases (19.4%), chlorinated solvents in 4 cases (11.1%), pesticides in 4 cases, and ionizing radiation exposure in 3 patients (8.3%). Our findings support the creation of a large-scale study to explore the impact of occupational exposures, particularly exposure to asbestos and hydrocarbons.

## 1. Introduction

Esophageal cancer is the sixth most common cause of cancer death worldwide [1]. In Europe, the overall 5-year patient survival rate was estimated to be 9.8% in 2012 [2]. Diagnosis and treatment improvements have been made but esophageal cancer remains highly fatal and very little data is available about risk factors other than behavioral ones. The recent Europe’s Beating cancer plan [3] and the French ten-year cancer control strategy [4] have made it a priority to develop research to better understand cancers with poor prognosis such as esophageal cancer and to investigate the implication of occupational exposures.

The French department most seriously affected by esophageal cancer is Finistère, in the region of Brittany. During the period 2007–2016, there was an excess frequency of esophageal cancer of 34% for men and 18% for women in Brittany [5], but this was still greater in the department of Finistère, where increased incidence and mortality were +55% and +82%, respectively, for men. As a similar phenomenon is observed for the group of cancers including lip, mouth, and pharynx cancers, the consensual hypothesis is that they are related to high rates of alcohol overconsumption and smoking in Britany, both of which are established risk factors for these cancers [6]. However, worldwide, the highest burden of esophageal cancer is localized from China to Iran, where sex ratio and habitus are different than in France [7,8,9,10].

Case-control studies have explored the association between esophageal cancer and other risk factors, such as occupational exposures. The Spanish study by Santibanez et al. [11] published in 2008, included 185 patients with newly diagnosed esophageal cancer and 285 matched controls. Association with squamous or adenocarcinoma cancer was influenced by occupational exposure. The professions found to be associated with squamous cell esophageal cancer were waiters and bartenders, miners, shotfirers, and stone cutters. Those associated with adenocarcinoma were carpenters and joiners, animal producers and related workers, and those working in building and related jobs, such as electricians. Richiardi et al. investigated the association between occupational history and upper aerodigestive tract (UADT) cancer risk in the ARCAGE European case-control study. This study included 1851 patients with incident cancer of the oral cavity, oropharynx, hypopharynx, larynx, or esophagus and 1949 controls, between 2000 and 2005 [12]. Among men, they found increased risks among painters, bricklayers, workers employed in the erection of roof coverings and frames, reinforced concreters, dockers, and workers employed in road construction, general construction of buildings, and cargo handling. Among women, there was no clear evidence of increased risks of UADT cancer in association with occupations or industrial activities.

Brittany is a coastal region in the west of France where marine, agriculture, and food industry occupations are well developed, according to an analysis by INSEE published in 2014 [13]. Seafarers and fish farmers are 4.5 times more concentrated here than in the rest of France. More specifically, in the cities of Brest and Lorient, there are many military personnel and firefighters. These characteristics are associated with particular occupational exposures to carcinogenic agents, such as pesticides, mineral dust (including asbestos), polycyclic aromatic compounds, ionizing radiation, and solvents. Ionizing radiation is a known risk factor for esophageal cancer recognized by the International Agency for Research on Cancer (IARC) [6].

To examine the suspected link between esophageal cancer and some occupational exposures, the PROG study for professional risks and esophageal cancers, took a systematic approach using a survey of employment history to identify the occupational exposures of patients with esophageal cancer.

## 2. Materials and Methods

### 2.1. Population

The PROG study was a monocentric prospective descriptive study conducted from June 2020 to November 2020 at a Teaching Hospital in Brittany, in the west of France. All participants were informed of the objectives of the study and gave their informed consent. The research protocol was approved by the research ethics comity of Brest, referenced 29BRC20.0071.

Since the study’s purpose was exploratory and descriptive (the main objective was not based on any statistical inference), no calculation of the required number of patients was realized.

The mean number of newly diagnosed esophageal cancer cases in Brittany for the period 2007–2016 is estimated at 288 per year for men and 71 for women. We estimated that 36 patients from our hospital could be included in the study for a period of 6 months.

Eligible patients were adults, had been diagnosed as having esophageal cancer confirmed by histological analysis, and were being followed by the medical oncology team at our hospital. A total of 37 patients were identified, of which one patient declined to be interviewed. 

Firstly, a systematic review on occupational exposure and esophageal cancer was performed by the occupational diseases center physicians. Specific exposures discussed in previous studies were included for assessment by questionnaire. The questionnaire was also built by occupational physicians in the Brest Teaching Hospital and validated during a first interview. The final version was used during the study. A trained occupational medicine resident conducted a systematic individual interview with each subject during medical consultation and analyzed their medical file to gather data about demographic characteristics, way of life (including tobacco and alcohol consumption and diet), medical history, and characteristics of cancer. The employment history and occupational exposures were recorded in detail, and the subjects’ employments were kept in the study if they had lasted a minimum of 6 months and were classified according to the French nomenclature of professions and occupational categories of 2016 [14].

### 2.2. Statistical Analysis

All data was compiled in an Excel file. Data analysis was performed using XLSTAT software (Addinsoft, New York, NY, USA). The results for continuous variables are shown as medians with the interquartile range (IQRs). The data from each part of the questionnaire were ranked and presented as numbers and percentages. Due to the low number of participants, we were not able to make comparative tests. 

## 3. Results

Among the 36 participants, the majority (n = 21, 58.3%) presented an adenocarcinoma esophageal cancer, 13 (36.1%) presented squamous cell cancer, and 2 (5.6%) presented other types (a combination of adenosis and squamous cell cancer and a neuroendocrine tumor). Most of these cancers (n = 25, 69.4%) were localized in the lower third of the esophagus. 

In terms of demographic characteristics and habitus (Table 1), the patients were mostly men (n = 27, 75.0%), smokers (n = 25, 69.4%), and 16 (44.4%) were regular consumers of alcohol. The median age was 68.5 years and the age range of the cohort was between 42 to 84 years. The majority had a history of cardiovascular diseases (n = 20, 56.6%) and 12 (33.2%) patients were obese or overweight. For cancer characteristics and history, eight of the patients (22.2%) had another cancer, of whom two had digestive cancer, three (8.3%) patients had Barrett’s esophagus, and five (13.6%) had a familial history of esophageal or gastric cancer. Most of the included esophageal cancer cases had adenocarcinoma histology (58.3%), followed by squamous cell carcinoma (36.1%), and other types (5.6%). Most cancers were located in the lower esophagus. (69.4%) 

Table 2 presents the results of the data analysis of occupational exposures.

Asbestos exposure was traced, and we found that 11 patients (30.6%) had been highly exposed to handling asbestos materials without personal protective equipment at their jobs as electricians, mechanics, boilermakers, or farmers. The mean period of exposure was 12.6 years, and the mean latency was estimated as 23.7 years between the end of exposure and the diagnosis of esophageal cancer. Twenty other subjects (55.6%) declared an environmental exposure more specifically in workplaces built using asbestos.

The second most frequent type of exposure was to hydrocarbons and their derivatives. In our study, we observed that seven patients (19.4%) had regularly come into contact with benzene by handling fuel with occupational work tasks and sectors including mechanic and boilermaker, and used cutting oils derived from hydrocarbons for metal machining. Moreover, one patient was exposed to bitumen during his job as a construction worker laying asphalt, and four patients (11.1%) were exposed to aromatic amines by working with paint and in printing (in ship maintenance for three of them).

In the field of chemistry, two workers at plastic factories and two members of biochemical laboratories (11.1%) were exposed to aromatic amines and sulfuric acid. 

Four patients (11.1%) talked about their use of chlorinated solvents during cleaning and degreasing operations, with one of them using it in the textile industry.

Indeed, for other occupational exposures, four patients (11.1%) declared handling pesticides including two farmers with regular exposure. 

Exposure to radiation was noted in four cases (11.1%) and all of them were exposed in the military nuclear field. One patient was also exposed to electromagnetic radiation in the telecommunication field.

## 4. Discussion

As this is a preliminary and descriptive study, our approach has several limitations. The primary limitation is selection bias, as we included a small cohort of 36 patients over a limited period. We also must consider recall bias, as our results are based on interviewing the patients—most of whom were retired—rather than on actual measurements of their exposures. Through the years, our institution has developed expertise in occupational cancers and offer to a patient a dedicated consultation to explore their exposure to carcinogens at work. To limit the data collection bias, a review of the literature was performed to select the carcinogenic agents whose association with esophageal cancer is discussed. Ionizing radiation, pesticides, chlorinated solvents, benzene, polycyclic aromatic hydrocarbons, sulfuric acid, aromatic amines, and asbestos were chosen. We used questionnaires drawn up for the occupational cancer consultation, detailing the activities exposed to those carcinogens. The third limitation is the absence of a control group. These limitations limit our study’s conclusion as far as the occupational exposures highlighted and do not prove that their associations with esophageal cancer are causal. At present and to our knowledge, there is no systematic review of occupational risk factors and esophageal cancer. 

When looking more specifically at occupational exposure declared in the PROG study, asbestos appears prominent. Asbestos fibers are known to be a cause of mesothelioma, and cancers of the lung, larynx, and ovary. A positive association has also been observed with stomach and colorectal cancers, but the association of asbestos with esophageal cancer remains a matter of discussion [6,15]. The meta-analysis by Li and al. demonstrated a higher standardized mortality ratio (SMR) in favor of an increased risk of esophageal cancer for male workers highly exposed to asbestos [16]. In our study, almost a third of the cohort had been significantly exposed. Exposure to hydrocarbons and their derivatives was the second most frequent exposure observed here. In the scientific literature, the results of different studies are inconsistent [17]. However, a positive association has been described between occupational exposure to bitumen and its emissions and the upper aerodigestive tract including the esophagus [17,18,19,20]. A few patients in our cohort used chlorinated solvents for cleaning and degreasing work. The cancer risks among workers using chlorinated solvents, such as trichloroethylene, have been studied in Danish cohorts, but the association of exposure to these substances with esophageal cancer was not consistent [21,22,23] and potentially affected by statistical bias related to tobacco and alcohol use. Regarding ionizing radiation exposure, four patients were exposed by working in the military nuclear field. Their exposure was considered significant due to the status of category A worker, defined by the French Labor Code [24], meaning they were likely to receive an effective dose greater than 6 mSv or an equivalent dose greater than 150 mSv for the skin and/or the extremities, during twelve consecutive months. Three patients presented an adenocarcinoma, the last patient was suffering from a neuroendocrine carcinoma. Two patients have no history of smoking, alcohol abuse, or being overweight. Obesity and being overweight are known risk factors for esophageal adenocarcinoma [25,26]. In our study, over the 21 patients with adenocarcinoma, seven presented a body mass index ≥ 25. For the two of them who did not report smoking or alcohol abuse, no specific occupational exposure was described. None of the patients reported any specific eating habits. Frequent consumption of hot beverages and hot food might be a risk factor for esophageal cancer in Chinese and South American populations [27].

The search for meaning is a common response of patients seeking to cope with the stress of cancer, and meaning-centered psychotherapy has been proven to have benefits for spiritual well-being and quality of life in patients with advanced cancers [28]. Of the 37 eligible patients identified for our study, only one declined to participate. These patients were seeking dedicated time to discuss the courses of their lives. Their positive feedback is an encouraging sign for future research that will help us to understand other risk factors besides tobacco and alcohol.

## 5. Conclusions

In conclusion, the PROG study is a pilot study offering a systematic approach using a survey of employment history to identify the occupational exposures of patients with esophageal cancer. Our study underlines some specific occupational exposures such as asbestos, hydrocarbons and their derivatives, chlorinated solvents, and pesticides. Our findings support the creation of a large-scale study to explore the impact of occupational exposures, particularly those to asbestos and hydrocarbons.

## Figures and Tables

**Table 1 ijerph-19-09782-t001:** Demographics, life habits, and cancer characteristics.

Characteristics	Patients (n = 36)
**Sex**	
Male	27 (75.0%)
Female	9 (25.0%)
**Smoking status**	
Never	11 (30.6%)
Past and current	26 (69.4%)
**History of alcohol abuse**	
Absent	20 (55.6%)
Present	16 (44.4%)
**BMI * (kg/m^2^)**	
≤25	24 (76.8%)
>25	12 (33.2%)
**Barrett’s esophagus**	
Absent	33 (91.7%)
Present	3 (8.3%)
**Familial history of esophageal or gastric cancer**	
Absent	31 (86.4%)
Present	5 (13.6%)
**Histologic type**	
Squamous-cell carcinoma	13 (36.1%)
Adenocarcinoma	21 (58.3%)
Other types	2 (5.6%)
**Location**	
Lower	25 (69.4%)
Middle	9 (25%)
Upper	2 (5.6%)

* BMI = body mass index.

**Table 2 ijerph-19-09782-t002:** Data analysis of included patients’ occupational exposures.

Exposure	Patients (n = 36)
**Asbestos**	
Absent	5 (13.8%)
Environmental exposure	20 (55.6%)
High exposure	11 (30.6%)
**Hydrocarbons and their derivatives**	
Absent	23 (63.9%)
Present	13 (36.1%)
**Aromatic amines**	
Absent	30 (83.3%)
Present	6 (16.7%)
**Chlorinated solvents**	
Absent	32 (88.9%)
Present	4 (11.1%)
**Sulfuric acid**	
Absent	32 (88.9%)
Present	4 (11.1%)
**Pesticide**	
Absent	32 (88.9%)
Present	4 (11.1%)
**Ionizing radiation**	
Absent	33 (91.7%)
Present	4 (11.1%)

## Data Availability

Data are located in Brest Teaching Hospital, Unit of research and clinical investigation. Specific requests can be sent to the corresponding author.

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
