# Peer review of "Occupational Exposures and Esophageal Cancer: Prog Study"

_ijerph, 2022, doi:10.3390/ijerph19169782_

Round 1

Reviewer 1 Report

A few more corrections and additions should be made to the simple and straightforward manuscript:

Table 1: Especially for the two known main risk factors, smoking and alcohol consumption, only two categories are reported here. Despite the small number of cases, more details should be reported here (e.g. pack years).

Table 2: The term "passive exposure" seems unusual. This should be explained. What does quantitative mean compared to "high exposure".

For other hazards only two categories "absent" and "present" are given. With regard to retrospective lifetime exposure, this seems insufficient to me.

Line 82: What is meant by "XXX"?

Line 83: Usually we talk about "informed consent" and not "informed non-opposition". Please explain the difference here. If there is none, I suggest the change in wording. 

Line 84: Which ethics committee exactly reviewed the study? Only the reference number is not sufficient here.

Line 91: What do you mean by "Eligible patients were major, ...."?

Line 99: ".... occupational exposures were detailed, ...." Should this mean "were recorded in details"

Line 148: What is meant by "recruitment bias"? Is it the same as selection bias?

Line 150: What is meant by "declaration bias"? Recall bias could occur in the case of medical history.

Line 151: I would suggest to use the term "exposure" instead of "exposition" all over the manuscript

Line 189: What does "current or past" mean here?

Author Response

RESPONSES TO THE REVIEWERS: Reviewer 1

A few more corrections and additions should be made to the simple and straightforward manuscript:

Table 1: Especially for the two known main risk factors, smoking and alcohol consumption, only two categories are reported here. Despite the small number of cases, more details should be reported here (e.g. pack years).

Table 2: The term "passive exposure" seems unusual. This should be explained. What does quantitative mean compared to "high exposure".

For other hazards only two categories "absent" and "present" are given. With regard to retrospective lifetime exposure, this seems insufficient to me.

We thank the reviewer for his comments. Unfortunately, we don’t have more data to report. Regarding asbestos, we correct the term “passive exposure” by “environmental exposure”. It is a low exposure relating to the workplace. High exposure concerns the workers who handled asbestos products (dismantling, repair, maintenance) without personal protective equipment.

Line 82: What is meant by "XXX"?

It is the Teaching Hospital of Brest.

Line 83: Usually we talk about "informed consent" and not "informed non-opposition". Please explain the difference here. If there is none, I suggest the change in wording. 

We change in wording as suggest.

Line 84: Which ethics committee exactly reviewed the study? Only the reference number is not sufficient here.

It is the research ethics committee of Brest.

Line 91: What do you mean by "Eligible patients were major, ...."?

We mean adults (they have reached the age of the legal majority in France), the correction has been made.

Line 99: ".... occupational exposures were detailed, ...." Should this mean "were recorded in details"

Line 148: What is meant by "recruitment bias"? Is it the same as selection bias?

Line 150: What is meant by "declaration bias"? Recall bias could occur in the case of medical history.

We understand this comment, we mean indeed "were recorded in details", selection and recall bias.

Line 151: I would suggest to use the term "exposure" instead of "exposition" all over the manuscript

We correct the term as suggest.

Line 189: What does "current or past" mean here?

Some patients were still overweighted, some reached a normal weight due to the oesophageal cancer symptoms and the side effects of the treatments. We withdraw the mention “current or past” for more clarity.

Reviewer 2 Report

Dear Authors,

Thank you for an interesting manuscript. Overall, the manuscript is well arranged. However, the writing style is not as normal. Therefore, the manuscript should correct some minor points before publication. 

Minor's comments:

1. Abstract: Re-write the abstract with full sentences for easier understanding. Please consider withdrawing some terms such as "Background:", "Aim:",... which are abnormal in the abstract.

2. Table captions should be checked and made clearly described. (e.g. Table 1.: patients characteristics; Table 2. occupational exposures)

3. In the results and discussion sections, there are too many paragraphs and fewer sentences. Please modify these results and discussion more clear and easy for the reader to follow the manuscript.

Author Response

We thank the reviewer 2 for his comments. We have taken his remarks into account and have modified the document according to the reviewer's recommendations.

Round 2

Reviewer 1 Report

The manuscript was significantly improved and revised following the recommendations of the reviewers. Nevertheless, the manuscript needs another minor revision.

For completeness, it should be mentioned that frequent consumption of hot beverages and hot food might be a risk factor for oesophageal cancer in Chinese and South American populations. (e.g. Chen, Y., Tong, Y., Yang, C. et al. Consumption of hot beverages and foods and the risk of esophageal cancer: a meta-analysis of observational studies. BMC Cancer 15, 449 (2015). https://doi.org/10.1186/s12885-015-1185-1)

Information on the age of the participants is still missing, but is important as age is a general risk factor for most diseases - especially also for cancer. Here, it would be sufficient to simply report the median and age range of the cohort in one sentence in the results chapter.

Furthermore, the manuscript should be checked pain stacking for spelling mistakes, errors in capitalization or grammar.

e.g.

Line 127: Wording proposal: “Most of the included esophageal cancer cases had adenocarcinoma histology (58.3%), followed by squamous cell carcinoma (36.1%) and other types (5.6%). Most cancers were located in the lower esophagus (69.4%).”

Line 133: “Table 2: Data ….”

Line 153:“Exposure to radiation was noted in four cases (11,1%) all of them were exposed ….”

Author Response

We are grateful for the reviewer's recommendations.

We mentioned the hot beverages and food in the discussion (line 199).

The age is now reported in the results (line 123).

We corrected the text as suggested.